# Density-Based Separation of Microbial Functional Groups in Activated Sludge

**DOI:** 10.3390/ijerph17010376

**Published:** 2020-01-06

**Authors:** Lin Li, Yaqi You, Krishna Pagilla

**Affiliations:** Department of Civil and Environmental Engineering, University of Nevada, Reno, NV 89557, USA; lli234@nevada.unr.edu (L.L.); you.yaqi@gmail.com (Y.Y.)

**Keywords:** activated sludge, density-based separation, enrichment of microbial functional groups, ammonia-oxidizing bacteria (AOB), nitrite-oxidizing bacterial (NOB), phosphate accumulating organisms (PAOs), qPCR, FISH

## Abstract

Mechanistic understanding of how activated sludge (AS) solids density influences wastewater treatment processing is limited. Because microbial groups often generate and store intracellular inclusions during certain metabolic processes, it is hypothesized that some microorganisms, like polyphosphate-accumulating organisms (PAOs), would have higher biomass densities. The present study developed a density-based separation approach and applied it to suspended growth AS in two full-scale domestic water resource recovery facilities (WRRFs). Incorporating quantitative real-time PCR (qPCR) and fluorescence in situ hybridization (FISH) analyses, the research demonstrated the effectiveness of density-based separation in enriching key microbial functional groups, including ammonia-oxidizing bacteria (AOB), nitrite-oxidizing bacteria (NOB) and PAOs, by up to 90-fold in target biomass fractions. It was observed that WRRF process functionalities have significant influence on density-based enrichment, such that maximum enrichments were achieved in the sludge fraction denser than 1.036 g/cm^3^ for the enhanced biological phosphorus removal (EBPR) facility and in the sludge fraction lighter than 1.030 g/cm^3^ for the non-EBPR facility. Our results provide important information on the relationship between biomass density and enrichment of microbial functional groups in AS, contributing to future designs of enhanced biological treatment processes for improved AS settleability and performance.

## 1. Introduction

In suspended growth biological wastewater treatment, activated sludge (AS) is an important factor for the overall performance, influencing the efficiency of treatment process and subsequent effluent quality. In this study, we focus on the density of AS, which plays a pivotal role in two key design considerations in water resource recovery facilities (WRRFs): the settleability of biomass that directly affects solid-liquid separation in the final clarifier, and the nutrient removal/recovery efficiency. In order to achieve the two overall goals, microbial group selection is often needed. For a long time, metabolism-based microbial selection is dominant among WRRFs. Some other approaches have been established to improve the settleability of AS, like the application of classifier and hydrocyclone in full-scale WRRFs [1,2]. On one hand, to achieve successful solid-liquid separation and avoid suspended solids being discharged in the effluent, sludge flocs must settle and compact well in clarifiers, a process determined by multiple factors including the density of AS [3]. Previous studies have confirmed a correlation between sludge settleability/settling velocity and biomass density, with potential mechanisms lying at the cell, floc, and process level [2,3,4,5]. Generally, dense and strong flocs are desired for good AS settling and compaction, while sludges containing high quantities of filaments have poor compressibility and settleability [6]. The quantity and quality of extracellular polymeric substances (EPS) plays an important role in AS flocculation, and loosely bound EPS could have a negative effect on bioflocculation and sludge-water separation [7,8,9]. Microbial community composition could also affect floc and settling properties [10]. On the other hand, to ensure compliance with stringent discharge limits, WRRFs must achieve enhanced nutrient removal and recovery [11]. Functional groups in AS, such as nitrifiers, denitrifiers, and polyphosphate accumulating organisms (PAOs), govern nutrient removal and often accumulate intracellular inclusions and thus have higher cell density compared to other microbial groups [5,12,13,14]. Therefore, selecting functional microbial groups in heavier biomass can kill two birds with one stone. Several studies have been done on how average solids density is related to bacteria in AS [5,12,15], however, floc is the basic unit that exists inside biological reactors and little is known on how flocs density affects microbial ecophysiology.

Despite all the implications of AS for suspended growth biological wastewater treatment, current understanding of AS performance at the microbial community level is still incomplete [9]. In particular, relationship between microbial community assembly and activated biomass density and sludge bioactivity is yet to be fully addressed. Global studies on WRRF AS have revealed complicated microbial community structure with varying abundance of functional groups such as ammonia-oxidizing bacteria (AOB), nitrite-oxidizing bacteria (NOB), and PAOs [16,17]. Microbial functional groups often generate and store intracellular inclusions during metabolic processes; therefore, their relative abundance and spatial distribution within AS may result in varying bulk biomass density and spatial heterogeneity of density within the same biomass. This feature can potentially be leveraged for better manipulating key microbial functional groups towards effective and efficient biological wastewater treatment.

Previous studies focused on single-cell density in wastewater treatment process [5]. This is the first study to investigate biomass density on both the cell level and the floc level, with an aim to provide insights for process effects and control. First, the research established a density-based biomass separation method for enhanced biological phosphorus removal (EBPR) and non-EBPR AS and applied quantitative real-time PCR (qPCR) to quantify four target functional groups in different density layers. Second, fluorescence in situ hybridization (FISH) was employed to examine how the morphology of flocs influenced density distribution. Finally, we used multivariate analysis to study the influence of WRRF operational conditions on AS solids density and functional group distribution. We demonstrated the feasibility of density-based separation of biomass and the utility of this approach in selecting and enriching microbial functional groups. Overall, this research contributes to a critical knowledge gap at the intersection of AS microbial community, floc morphology, biomass density, and WRRF biological performance.

## 2. Materials and Methods 

### 2.1. WRRFs, Sampling and Sample Preservation

AS samples were collected from two full-scale domestic WRRFs with different secondary treatment processes (Table 1). The Truckee Meadows Water Reclamation Facility (TMWRF, Reno, NV, USA) employs EBPR and serves the central Truckee Meadows region in northern Nevada. AS samples were collected from an aeration tank of the AS process at TMWRF. The South Truckee Meadows Water Reclamation Facility (STMWRF, Washoe County, NV, USA) employs conventional nitrification-denitrification treatment and supplies reclaimed water to irrigate landscaping, sports fields, and golf courses. AS samples were collected from an oxidation ditch with nitrogen removal function at STMWRF. Both facilities treat predominantly domestic municipal wastewater. All the collected activated samples were stored in 2 L sterilized high-density polyethylene bottles in the field and kept on ice during transportation (less than 30 min). Upon arrival at the laboratory, each sample was thoroughly mixed, and triplicate 2 mL aliquots were taken from the sample, rapidly frozen and kept at −20 °C before DNA extraction (no longer than two days). The remaining fresh samples were used for density separation test on the same day of sampling. 

### 2.2. Density-Based Separation of Biomass

Isopycnic centrifugation has previously been applied successfully to “clean” samples like pure cultures of microorganisms or cells and suspensions of subcellular particles [4,5]. However, AS is a complex of microbial biomass, debris of dead microorganisms, and organic and inorganic particles. These compositions are all enmeshed in flocs rich in EPS in suspended growth biological systems [12]. In this study, we applied density-based separation to AS of full-scale domestic WRRFs using the Percoll medium. Percoll (Sigma-Aldrich, St. Louis, MO, USA) is a low-osmotic-pressure medium with a density of 1.13 ± 0.05 g/cm^3^. It has been widely used in cell separation and microorganism density measurement [5,12,14,19,20]. It consists of homogeneously distributed silicon particles (with the density of 2.2 g/cm^3^ and the size of 15–30 nm). Those particles can be homogeneously suspended in 0.15 mM NaCl solution, which can maintain appropriate osmotic pressure for cells. Here we used supernatant from fresh AS to prepare homogeneous density suspensions, in order to keep osmotic pressure as much similar to that of AS biomass as possible. A series of homogeneous density suspensions were prepared using low-osmotic-pressure Percoll colloidal particles according to the method developed by Jang and Schular [21], with some modifications. Briefly, Percoll colloidal particles were mixed with AS supernatants collected in the sample concentration step in varying ratios to reach a total volume of 100 cm^3^ (Table 2). The net density of each density suspension was calculated by Equation (1):
(1)$$\rho_{net}=\frac{V_{sup}\times \rho_{sup}+V_{pc}\times \rho_{pc}}{100\;\mathrm{mL}}$$
where *ρ_net_* is the net density of suspension a, b and c (g/cm^3^); *V_sup_* is the volume of AS supernatant (cm^3^); *ρ_sup_* is the density of AS supernatant (1.04 g/cm^3^ herein); *V_Pc_* is the volume of Percoll colloidal particles (cm^3^); and *ρ_Pc_* is the density of Percoll colloidal particles (1.13 g/cm^3^ herein). 

To separate biomass, 10 mL of a Percoll suspension (a, b, or c in Table 2) at a fixed density and 2 mL of a condensed AS sample was added into a 15 mL sterile centrifuge tube, and the tube was gently shaken for homogenization. The tube was then centrifuged at 2000× *g* for 10 min, during which solids were separated into two layers. The top layer contained sludge biomass lighter than the used density suspension; the bottom layer contained sludge biomass denser than the used density suspension. A visualization of this separation process using three density suspensions is shown in Figure 1A. The separated solids were allowed to stabilize at 4 °C for 30 min, after which biomass in the top layer was transferred into a new sterile centrifuge tube by careful pipetting. Biomass in the bottom layer was then transferred to a second new sterile centrifuge tube. When dissolved solids were occasionally introduced, three washings with ultrapure water (Quality Biological, Inc., Gaithersburg, MD, USA) were performed and Percoll colloidal particles were subsequently removed. For each AS sample, triplicate homogeneous separations were conducted, and density distribution was calculated based on TS mass in each layer. Separated sludge biomass was washed by sterile phosphate-buffered saline before downstream biomolecular analysis. Details about AS pretreatment and characterization are provided in the Appendix A.

### 2.3. Total DNA Extraction and Quantitative Real-Time PCR

For each sample, total environmental DNA was extracted in triplicate using PowerViral Environmental RNA/DNA Isolation kit (Mo Bio Laboratories, Inc., Carlsbad, CA, USA) according to the manufacturer’s protocol. Triplicate extractions from each sample were pooled together.

Extraction yield and DNA quality were evaluated by NanoDrop One UV-Vis Spectrophotometer (Thermo Scientific, Madison, WI, USA), and subjected to conventional PCR for inhibition mitigation and PCR cloning. After sequencing verification, the cloned PCR products were used to generate calibration curves for qPCR. Details about conventional PCR, PCR cloning, sequencing, and sequence analysis are described in Appendix A.

qPCR has been widely applied in the quantification of microbial groups in environmental samples [22]. Here we conducted qPCR on BioRad CFX96 Touch Real-Time PCR Detection System (BioRad, Hercules, CA, USA) as before [23] and in accordance with the MIQE guidelines [24]. Typical reaction mixtures contained 1× SsoAdvanced Universal SYBR Green Supermix (Bio Rad), 2 μL of the environmental DNA template, and primers (0.48 μM each) in a total reaction volume of 20 μL. The thermal program consisted of initial denaturation at 95 °C for 3 min, followed by 40 cycles of 10 s denaturation at 95 °C, 20 s at the annealing temperature, and 30 s extension at 75 °C. Details of qPCR conditions in this study show in Table 3. Threshold cycle (C_t_) was determined using the default algorithm in CFX Manager Software (BioRad). Positive and negative controls were included in each run. A melting curve analysis was conducted after each run to verify reaction specificity [25]. Specificity was confirmed by gel electrophoresis and sequencing of randomly selected qPCR products. Assay limits and relative abundance calculation are detailed in Appendix A. For each analyzed microbial functional group, an enrichment factor was calculated as the fold change in its relative abundance: relative abundance in separated sample/relative abundance in unseparated sample.

### 2.4. Fluorescence In Situ Hybridization 

FISH was employed to obtain spatial information of microbial functional groups as well as their ecophysiology within sludge biomass. Fresh biomass samples were fixed in a newly prepared paraformaldehyde solution. In situ hybridizations were conducted according to Nielsen et al. (2009) [32] with minor modifications. Table 4 lists sequences, targets, hybridization conditions, fluorochromes, and references for the rRNA-targeted oligonucleotide probes (IDT, Coralville, IA, USA) used in this study. Samples on each slide were hybridized with 5 pmol each probe in 98 μL hybridization buffer (0.9 M NaCl, 20 mM Tris hydrochloride (pH 7.2), 0.01% sodium dodecyl sulfate, and formamide at the concentrations shown in Table 4) at 42 °C for ≤16 hours. Microscopic examinations were conducted on a confocal laser scanning microscopy (CLSM) (Leica TCS SP8) with a 63× oil immersion objective and excitation wavelengths of 488 nm and 543 nm. All image processing and analysis were performed with the standard software package provided by Leica. Additional image processing was performed using ImageJ [33]. 

### 2.5. Statistical Analysis

Statistical analysis was performed with R (3.2.4) (Boston, MA, USA) and vegan package (Oulu, Finland) [34]. Copy numbers were log-transformed when necessary to normalize the distribution and to achieve homogeneity of variance. Data were compared using the Student’s *t*-test or Welch’s *t*-test and using the Mann-Whitney *U* test in consideration of small sample sizes. Correlation analysis was performed to calculate the Spearman rank correlation coefficient between the dependent and independent variables. Multivariate analysis was conducted to investigate effects of WRRF operational conditions and AS characteristics (parameters in Table 1, DNA per gram of MLVSS, and total bacteria per gram of MLVSS) on density-based separation and enrichment results. We first conducted a stepwise model selection on variables using the function “ordistep” in the vegan R package, and then conducted distance-based redundancy analysis (db-RDA) with the retained factors. The relative contribution of each factor on the enrichment of each microbial functional group was determined through Permutational Multivariate Analysis of Variance (PERMANOVA) implemented by the “adonis2” function in the vegan R package.

### 2.6. Nucleotide Sequence Numbers 

Nucleotide sequences have been deposited in the GenBank under the accession numbers MK253254-MK253257, MK332240-MK332246, MK332249, MK332330-MK332331, and MK332336-MK332338 (16S rRNA genes) and MK333402-MK333405 (*amoA* genes).

## 3. Results

### 3.1. Density-Based Separation of Biomass

Using three homogeneous density suspensions consisting of low-osmotic-pressure Percoll colloidal particles at different ratios, AS samples were successfully separated into different density fractions (Figure 1A). A total of six subsamples were obtained for each condensed AS sample, including two fractions lighter or heavier than 1.030 g/cm^3^ (designated throughout the study as 1030T and 1030B, respectively; T and B representing top and bottom), two fractions lighter or heavier than 1.036 g/cm^3^ (1036T and 1036B, respectively), and two fractions lighter or heavier than 1.042 g/cm^3^ (1042T and 1042B, respectively). In general, the majority of AS biomass had a density range of 1.030–1.042 g/cm^3^ (Figure 1B), with TMWRF sludge density being 1.032 g/cm^3^ and STMWRF sludge density being 1.037 g/cm^3^ according to the method by Jang and Schulaer [21]. 

Environmental DNA was extracted from separated sludge fractions. In many cases, separated sludge fractions yielded higher concentrations of DNA than bulk sludge. This is particularly the case for STMWRF sludge. For example, compared to a DNA yield of 26.66 ± 0.01 μg/mL for unseparated STMWRF sludge, the 1030T and 1030B fraction of sludge yielded 170.47 ± 0.45 μg/mL and 73.79 ± 0.24 μg/mL DNA, respectively. In line with enhanced DNA yields from separated sludge fractions, higher bacterial concentrations, indicated by eubacterial 16S rRNA gene copy number, were detected in all the separated fractions. 

### 3.2. Microbial Groups in AS of Two Different Domestic WRRFs

A variety of bacterial groups were identified in both WRRFs, including AOB, *Ca*. Accumulibacter, *Nitrobacter* spp. and *Nitrospira* spp., as well as groups closely related to *Albidiferax* spp., *Flavobacterium* spp., *Methylophilus* spp., *Tolumonas* spp. and *Rhodoferax* spp. (Appendix A). Comparing the relative abundance of these microbial groups, AOB were 6.3 times more prevalent in the oxidation ditch of STMWRF than in the aeration tank of TMWRF, representing (4.8 ± 1.2) × 10^−4^ and (7.6 ± 2.5) × 10^−5^ of total eubacteria, respectively (*p* < 0.0005). With respect to NOB groups, the relative abundance of *Nitrobacter* spp. was similar in these two WRRFs, representing 0.62%−0.65% of total eubacteria (*p* = 0.696). The relative abundance of *Nitrospira* spp. was 42 times higher in STMWRF than in TMWRF, accounting for (3.9 ± 0.5) × 10^−3^ and (9.3 ± 3.3) × 10^−5^ of total eubacteria, respectively (*p* < 0.0005). PAOs were 4.2 time more prevalent in TMWRF than in STMWRF, representing 0.71% ± 0.07% and 0.17% ± 0.03% of total bacteria, respectively (*p* < 0.0005). Microscopic imaging further showed that AS flocs from TMWRF were significantly smaller than those from STMWRF (40 µm vs. 60 µm) (*p* < 0.005) (Appendix A). In addition, AS from STMWRF had more filamentous microbes and a relatively loose structure compared to TMWRF sludge. In contrast, TMWRF sludge showed round, compact flocs, and microbes there tended to cluster together (Appendix A).

### 3.3. Abundance of Target Microbial Groups after Density-based Separation

For AS collected from the aeration tank of TMWRF, separation with the density of 1.030 g/cm^3^, 1.036 g/cm^3^ and 1.042 g/cm^3^ all enriched AOB, NOB and PAOs (Figure 2). Similar results were observed in AS collected from the oxidation ditch of STMWRF. Maximum enrichment of individual microbial group was achieved under varying densities. For TMWRF, the sludge fraction denser than 1.036 g/cm^3^ had the best enrichment effect whereas for STMWRF, the sludge fraction lighter than 1.030 g/cm^3^ showed the maximum enrichment (Figure 2). 

#### 3.3.1. AOB Populations

Density-based separation significantly enriched AOB in sludge fractions (*p* < 0.0005), except in the 1042T fraction of TMWRF sludge (Figure 2). For TMWRF sludge, the highest enrichment was seen in the 1036B fraction (relative abundance (3.3 ± 0.1) × 10^−5^, enrichment factor 43). For STMWRF sludge, the highest enrichment was seen in the 1030T fraction (relative abundance (4.3 ± 0.4) × 10^−4^, enrichment factor 90). FISH analysis supported these qPCR-based observations and further indicated the spatial distribution and ecophysiology of enriched AOB in separated sludge flocs (Figure 3). In TMWRF sludge fractions, large AOB cells were embedded in compact flocs, while in STMWRF sludge fractions, small AOB were residing in loose flocs.

#### 3.3.2. *Nitrobacter* spp. NOB Populations

Separation significantly enriched *Nitrobacter* in sludge fractions (*p* < 0.0005), except in the 1036T fraction of STMWRF sludge (Figure 2). For TMWRF sludge, the highest enrichment was found in the 1036B fraction (relative abundance 12.25% ± 0.89%, enrichment factor 20). For STWMRF sludge, the highest enrichment was found in the 1030T fraction (relative abundance 27.56% ± 5.5%, enrichment factor 43) (*p* < 0.0005). Moreover, *Nitrobacter* spp. tended to accumulate in denser fractions than in lighter fractions for TMWRF sludge, whereas there was no such trend for STMWRF sludge (Figure 2). FISH images further showed that *Nitrobacter* cells in TMWRF sludge were generally larger than those in STMWRF sludge (Figure 3).

#### 3.3.3. *Nitrospira* spp. NOB Populations

Separation significantly enriched *Nitrospira* NOB (*p* < 0.0005), except in the 1042T fraction of TMWRF sludge (Figure 2). For TMWRF sludge, the greatest enrichment was seen in the 1036B fraction (relative abundance (4.0 ± 0.2) × 10^−4^, enrichment factor 4). For STMWRF sludge, the 1030T fraction had the greatest enrichment (relative abundance 7.6% ± 1.1%, enrichment factor 19). Similar to *Nitrobacter*, *Nitrospira* tended to accumulate in denser fractions than in lighter fractions of TMWRF sludge, whereas there was no such trend for STMWRF sludge (Figure 2). However, unlike AOB and *Nitrobacter* NOB, *Nitrospira* cells in the two WRRFs did not significantly differ in size (Figure 3). 

#### 3.3.4. PAOs Populations

Separation significantly enriched PAOs (*p* < 0.0005), except in the 1030T fraction of TMWRF sludge (Figure 2). The highest enrichment was found in the 1036B fraction of TMWRF sludge (relative abundance 9.1% ± 0.4%, enrichment factor 13) and the 1030T fraction of STMWRF sludge (relative abundance 9.8% ± 1.3%, enrichment factor 59). Similar to *Nitrobacter* and *Nitrospira*, PAOs tended to accumulate in denser fractions than in lighter fractions of TMWRF sludge; there was no such trend for STMWRF sludge (Figure 2). Consistently, FISH analysis showed agglomeration of PAOs in sludge flocs after separation (Figure 3). Similar to AOB and *Nitrobacter* spp., cells of PAOs were larger in TMWRF sludge than in STMWRF sludge (Figure 3). 

## 4. Discussion

### 4.1. AS Density Distribution in Two WRRFs

The measured densities of AS in the two domestic WRRFs were consistent with previously reported density range of AS [39,40]. As shown by Figure 1B, AS in EBPR WRRF TMWRF had relatively uniform density distribution, whereas non EBPR WRRF STMWRF sludge had a wider density distribution. These differences could be partially attributed to the presence of larger, looser flocs and more filamentous microbes in STMWRF sludge (Appendix A). Flocs are highly heterogeneous, containing inert materials as dense as 1.24 g/cm^3^ [40]; thus sludge with more flocs may have a wider density distribution. Filamentous microbes, along with EPS matrix, could facilitate floc formation [2,41,42]. Comparing bulk and separated sludge fractions, we observed enhanced DNA extraction efficiency in separated AS fractions, likely due to floc disintegration and subsequent release of microbial cells and extracellular DNA from flocs [22,43]. Additionally, Percoll colloid particles that are coated with a water-soluble polymer may have contributed to more complete DNA recovery and removal of PCR inhibitory substances [44,45].

### 4.2. Enrichment of Microbial Groups by Density-Based Separation

The microbial groups identified in AS of the two WRRFs are part of the core community of abundant organisms commonly found in WWRFs [46]. In general, the abundance of AOB, NOB, and PAOs observed here was similar to others [47,48,49]. Moreover, *Nitrobacter* NOB were more prevalent than *Nitrospira* NOB in this study, likely due to faster growth of *Nitrobacter* than *Nitrospira* under higher nitrite levels [50]. It should be noted that *Nitrospira* is the most diverse NOB genus and forms a deeply branching lineage in the *Nitrospirae* phylum [49,50,51]. Thus, the true abundance of all *Nitrospira* populations may have been underestimated. Overall, we observed higher prevalence of AOB and NOB in STMWRF while more PAOs in TMWRF, consistent with the functions of these two WRRFs.

Density of single bacterial cells and morphology of flocs can be two critical reasons behind the enrichment of target microbial groups in separated AS biomass fractions. For instance, Winkler et al. (2013) [5] previously found that *Nitrobacter winogradskyi*, a typical NOB species, had a density of 1.108 ± 0.007 g/cm^3^ in reactors. Thus, accumulation of *Nitrobacter* spp. in denser fractions of TMWRF sludge was expected. However, enrichment of *Nitrobacter* spp. in the 1030T fraction of STMWRF sludge was unexpected. Since DNA extraction efficiency and PCR inhibition had been considered in the relative abundance calculation, this difference between TMWRF and STMWRF sludge in highest enrichment fractions was most likely attributed to AS characteristics including floc structure and/or microbial community assembly. For example, total bacterial abundance was highest in the 1030T fraction for STMWRF, and flocs from STMWRF were larger and looser. Differential ecophysiology of microbial functional groups in the two WRRFs could be another reason. As shown by FISH analysis (Figure 3), *Nitrobacter* cells in TMWRF sludge were generally larger than those in STMWRF sludge, indicating a higher density of *Nitrobacter* cells in TMWRF sludge and thus easier accumulation in denser fractions. During EBPR process, PAOs aerobically accumulate phosphorus inside their cells in the form of polyphosphate to energize anaerobic carbon uptake [13,28,47,52]. This could lead to larger, heavier PAO cells containing inclusions and thus the enrichment of such cells in denser sludge fractions, as seen here for TMWRF sludge. Ecophysiology of PAOs likely differed in the EBPR TMWRF and the non-EBPR STMWRF such that cells in STMWRF sludge were smaller in size (Figure 3). This factor, along with floc characteristics discussed above, may have led to the highest enrichment of PAOs in the 1030T fraction for STMWRF. It should be noted that our analysis focused on *Ca*. Accumulibacter. PAOs from diverse lineages are actively present in EBPR WRRFs, and some of them may also be present in TMWRF sludge [48]. Future research should look at the abundance and enrichment of PAOs other than *Ca*. Accumulibacter for a more comprehensive assessment. 

Interestingly, the densest sludge fractions (*ρ* > 1.042 g/cm^3^) did not yield the maximum enrichment of any tested microbial functional group, which may be due to the dominance of inert materials rather than microbes in those sludge proportions. AS has highly complicated composition and heterogeneous structure, with flocs being the basic unit. Firm and compact flocs tend to have higher density and are difficult to deflocculate [2]. While we used slow-speed centrifuge for biomass separation with the intention to maintain original floc structure, it is possible that this procedure had slightly modified floc structure and thus contributed to the overall enrichment effects. Additionally, EPS physicochemistry, particularly polymerization degree of proteinaceous substrates, has large impact on EPS functionality [53,54], which in turn influences floc stability and AS flocculation [7,8,9]. Future research should address the impact of EPS on density-based enrichment of microbial groups. 

### 4.3. WRRF Operational Conditions Affecting Density-Based Enrichment of Microbial Functional Groups

In order to explore practical factors that may influence and/or predict density-based enrichment of microbial functional groups, we performed multivariate analysis. Distance-based RDA (db-RDA) clearly showed that enrichment of AS microbial functional groups in TMWRF and STMWRF was largely distinct (Figure 4). 

WRRF operational condition solids retention time (SRT) explained variations in the observed enrichment effects (*R*^2^*_Adonis_* = 0.126, *P_Adonis_* = 0.085). Furthermore, DNA amount per gram of MLVSS (mixed liquid volatile suspended solids), rather than MLVSS (indicator of suspended biomass) itself, significantly explained variations in the observed enrichment effects (*R*^2^*_Adonis_* = 0.302, *P_Adonis_* = 0.003). Specifically, strong correlation was observed between DNA amount per gram of MLVSS and the enrichment of *Nitrobacter* NOB for both WRRFs (Spearman *r* = 0.895, *p* < 0.001) (Appendix A). Extracellular DNA, secreted by live cells or by lysis of cells, is abundant in AS and constitutes an important structural component of sludge floc EPS [43,50,53]. Higher amount of extracellular DNA may suggest EPS-rich flocs that allowed better separation and enrichment of microbial functional groups. Future research should investigate the impact of EPS and other factors that can affect AS biomass separation and enrichment of microbial groups. A better understanding of sludge microbial community is also needed, as floc community assembly may influence density-based separation and enrichment, and vice versa. Future research should also verify density-based separation and enrichment at larger scales.

## 5. Conclusions

In this study, we demonstrated that density-based biomass separation could enrich AOB, NOB and PAOs from bulk AS of full-scale WRRFs up to 90-fold. We observed that for EBPR and non-EBPR WRRFs, maximum enrichment was achieved in different density fractions. In particular, for a full-scale domestic EBPR WRRF, the highest enrichment was achieved in the sludge fraction denser than 1.036 g/cm^3^; for a full-scale WRRF performing conventional biological removal of C and N, the highest enrichment was achieved in the sludge fraction lighter than 1.030 g/cm^3^. Microbial ecophysiology (large vs. small cell size), floc structure (compact vs. loose), sludge characteristic (e.g., DNA amount per gram of MLVSS), and operational condition (e.g., SRT) likely contributed to these differences. Overall, our results provide important information for future research on and design of enhanced biological treatment systems where microbial functional groups in AS could be preferentially recycled back to the bioreactor.

## Figures and Tables

**Figure 1 ijerph-17-00376-f001:**
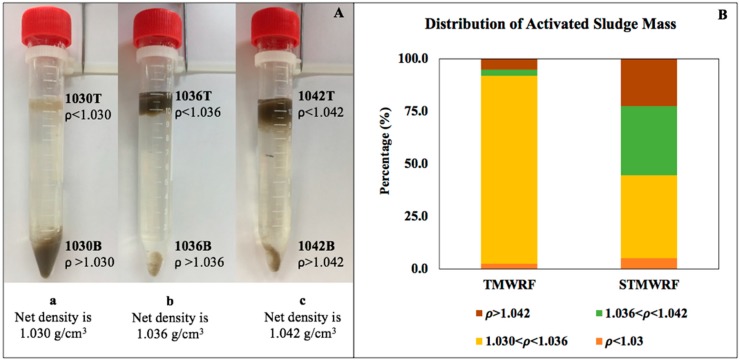
Density-based homogeneous separation of AS biomass. (**A**) A visualization of separation of AS from the EBPR facility TMWRF, using three density suspensions a, b and c as detailed in Table 1. AS was separated into two fractions, either lighter (e.g., 1030T) or denser (e.g., 1030B) than the net density of the suspension (e.g., 1.030 g/cm^3^ for the density suspension a). (**B**) Mass distribution of AS from an aeration tank in the EBPR facility TMWRF and from an oxidation ditch with nitrogen removal function in STMWRF performing conventional treatment.

**Figure 2 ijerph-17-00376-f002:**
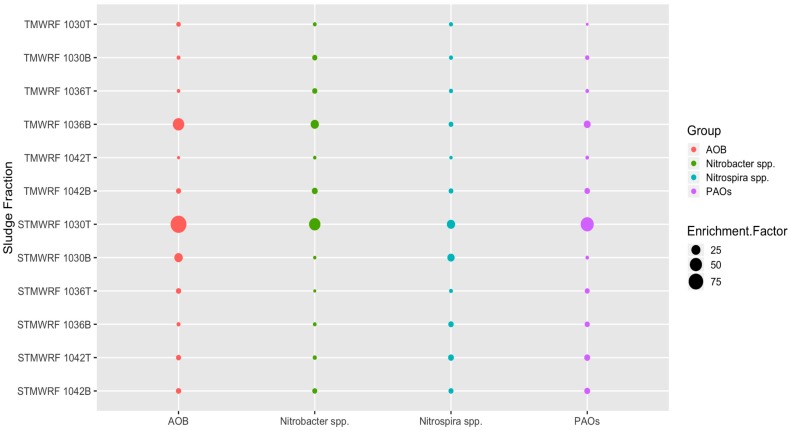
Enrichment factors for each microbial functional group achieved in sludge fractions separated using various homogenous densities. For example, 1030T designates the fraction lighter than 1.030 g/cm^3^ and 1030B designates the fraction denser than 1.030 g/cm^3^.

**Figure 3 ijerph-17-00376-f003:**
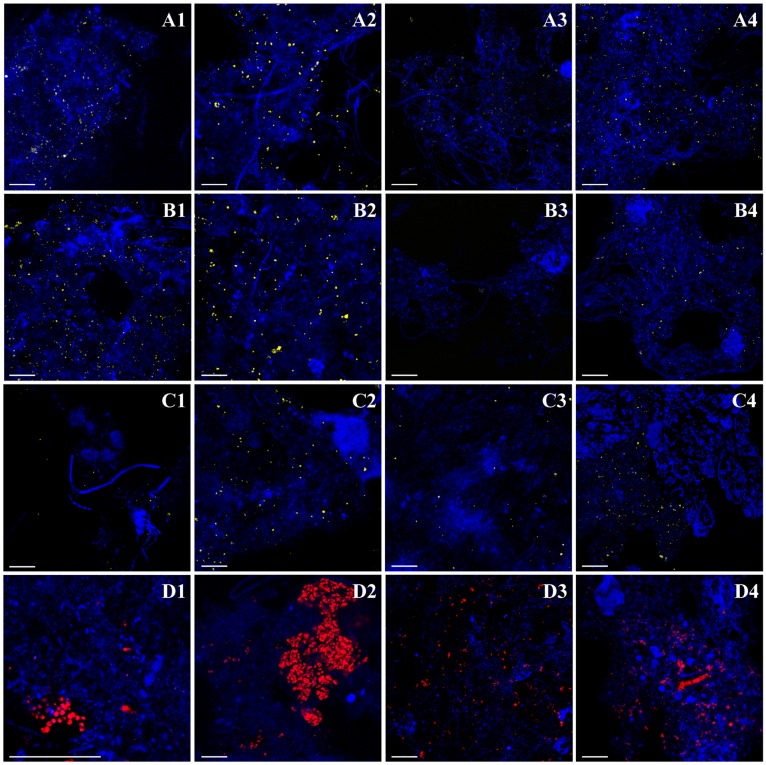
CLSM images showing morphology and spatial organization of AS containing (**A**) AOB (yellow), (**B**) *Nitrobacter* spp. NOB (yellow), (**C**) *Nitrospira* spp. NOB (yellow), and (**D**) PAOs (red), along with total eubacteria (blue). Shown here are (1) unseparated TMWRF sludge, (2) TMWRF sludge fraction with the best enrichment effects (denser than 1.036 g/cm^3^), (3) unseparated STMWRF sludge, (4) STMWRF sludge fraction with the best enrichment effects (lighter than 1.030 g/cm^3^). Scale bar indicates 20 µm.

**Figure 4 ijerph-17-00376-f004:**
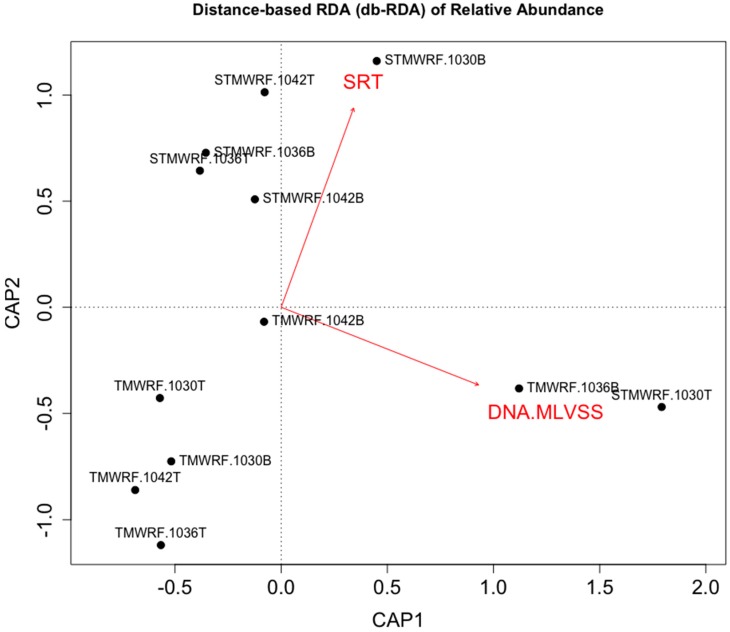
Distance-based redundancy analysis (db-RDA) of microbial enrichment achieved in various sludge fractions (constrained inertia = 42.76%). Enrichment factors were constrained by facility types (TMWRF employing EBPR vs. STMWRF performing conventional treatment). Both facility operational condition (SRT) and sludge characteristic (DNA amount per gram of MLVSS) explained variations in enrichment (SRT: *R*^2^*_Adonis_* = 0.126, *P_Adonis_* = 0.085; DNA/MLVSS: *R*^2^*_Adonis_* = 0.302, *P_Adonis_* = 0.003).

**Table 1 ijerph-17-00376-t001:** Description of the two full-scale domestic water resource recovery facilities and characteristics of activated sludge.

Parameters	TMWRF ^1^	STMWRF ^2^
Source of Wastewater	50% Domestic, 50% industrial	Mainly domestic
Biological Process ^2^	EBPR	C, N
Solids Retention Time (day)	~2.5	12 to 15
Floc size ^3^ (μm)	40	60
Filament Abundance ^4^	Some	Common
MLVSS (mg/L) ^5^	31.0	32.5
Influent Flow Rate (×1000 m^3^/day)	141	15
Influent BOD ^6^ (mg/L)	250	330
Effluent BOD (mg/L)	~5	7
Effluent Total N (mg/L)	0.2	8.4
Effluent Total P (mg/L)	0.4	2.1
Effluent TSS ^7^ (mg/L)	2.6	<5

^1^ TMWRF, Truckee Meadows Water Reclamation Facility (Reno, NV, USA); STMWRF, South Truckee Meadows Water Reclamation Facility (Washoe County, NV, USA); ^2^ The biological processes are classified as carbon (C) removal, nitrogen (N) removal, and phosphate (P) removal; EBPR, enhanced biological phosphorus removal; ^3^ Floc sizes were determined as the average of 20 randomly selected flocs; ^4^ Filament abundance was determined by six scales (few, some, common, very common, abundant, and excessive) according to a previous study [18]; ^5^ MLVSS, mixed liquid volatile suspended solids; ^6^ BOD, biochemical oxygen demand; ^7^ TSS, total suspended solids.

**Table 2 ijerph-17-00376-t002:** Density Composition of Suspensions.

Suspension Label	Net Density (g/cm^3^)	Supernatant of AS After Centrifuge (cm^3^)	Percoll (cm^3^)	Total Volume (cm^3^)
a	1.030	79.4	20.6	100
b	1.036	74.6	25.4	100
c	1.042	69.8	30.2	100

**Table 3 ijerph-17-00376-t003:** Primers Used in This Study and qPCR Performance.

Target Gene	Primer	Sequence (5′-3′)	Annealing Temp (°C)	Length (bp)	qPCR Performance		Reference
					R^2^	Efficiency	
Eubacterial16S rRNA gene ^1^	27F	AGAGTTTGATCMTGGCTCAG	60	~1500	--	--	[26]
	1492R	GGWTACCTTGTTACGACTT					
Eubacterial 16S rRNA gene	1369F	CGGTGAATACGTTCYCGG	60	124	0.9966	91.09%	[27]
	1492R	GGWTACCTTGTTACGACTT					
PAOs 16S rRNA gene	518F	CCAGCAGCCGCGGTAAT	65	351	0.9985	95.56%	[28]
	846R	GTTAGCTACGGCACTAAAAGG					
*Nitrosomonas* spp. *amoA* gene	amoA-1F	GGGGTTTCTACTGGTGGT	60	491	0.9990	91.62%	[29]
	amoA-2R	CCCCTCKGSAAAGCCTTCTTC					
*Nitrospira* spp. 16S rRNA gene	NSR1113F	CCTGCTTTCAGTTGCTACCG	60	150	0.9994	93.82%	[30]
	NSR1264R	GTTTGCAGCGCTTTGTACCG					
*Nitrobacter* spp. 16S rRNA gene	Nitro119F	ACCCCTAGCAAATCTCAAAAAACCG	60	227	0.9994	92.40%	[31]
	Nitro1423R	CTTCACCCCAGTCGCTGACC					

^1^ Used in conventional PCR to generate nearly complete 16S rRNA gene for cloning.

**Table 4 ijerph-17-00376-t004:** Oligonucleotides Used for FISH.

Target Prokaryote	Probe	Sequence (5′-3′)	Formamide (%)	Fluorochrome	Reference
Eubacteria	EUB 338	GCTGCCTCCCGTAGGAGT	35	ALEX488	[32]
β-Proteobacterial AOB ^1^	Nso1225	CGCCATTGTATTACGTGTGA	35	CY3	[35]
Genus *Nitrospira* (sublineage 1 and 2) ^2^	Ntspa1026	AGCACGCTGGTATTGCTA	20	CY3	[36]
Genus *Nitrobacter*	NIT3	CCTGTGCTCCATGCTCCG	40	CY3	[37]
*Candidatus* Accumulibacter	PAOmix	PAO462, PAO651 and PAO846	35	CY5	[38]
	PAO462	CCGTCATCTACWCAGGGTTTAAC	35	CY5	
	PAO651	CCCTCTGCCAAACTCCAG	35	CY5	
	PAO846	GTTAGCTACGGCACTAAAAGG	35	CY5	

^1^ AOB, ammonia-oxidizing bacteria; ^2^ Competitor probe: CCTGTGCTCCAGGCTCCG.

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
