# Peer review of "Density-Based Separation of Microbial Functional Groups in Activated Sludge"

_ijerph, 2020, doi:10.3390/ijerph17010376_

Round 1

Reviewer 1 Report

In this manuscript, the authors attempted to investigated the biomass density on both cell level and floc level, and hence provide the overall guidance based for process effects and control. Firstly, the author was established biomass density separation method, and then applied qPCR to quantify four target functional groups in different density layers; combining with FISH, the morphology of flocs influences density distribution. Finally, used ordination analysis to study the influence of operational conditions in different WRRFs on AS solids density. Overall, this study fit the aim and scope of ijerph, and can lead to good audience in the field of wastewater engineering.

The specific comments were:

Flocculation is important for solid-liquid separation of activated sludge and should be mentioned in the introduction. In section 2.3. What is the housekeeping gene for Q-PCR in this study? How to ensure the uniformity of Q-PCR results? Line 196. “by what”? Line 196-199, 215-217, et al. Please move the unrelated part to the section discussion. Line 225-230. Please show specific data. In figure 3. Why the scale bar is not uniform. In figure 4. Figure 4 is incomplete. The section conclusion needs to be further streamlined, only highlighting the main conclusion. Line 360-367, et al. Move the outlook of this research to the section discussion. In figure S1. Species names need to be italicized.

Author Response

ijerph-657007

The International Journal of Environmental Research and Public Health Journal

Dear Editor,

We thank you and the reviewers for taking the time to consider and review our manuscript entitled “Density-based Separation of Microbial Functional Groups in Activated Sludge”. In response to reviewer comments and suggestions, we have made major revisions to the manuscript. We have improved the overall quality of the manuscript by reorganizing the main text, extending introduction and discussion, adding necessary and recent citations, and providing method details in the supplementary materials. We also improved grammar and language. We hope these changes have made the manuscript more appropriate for publication in the IJERPH journal.

Our specific, point-by-point responses to reviewer comments are provided below. Please note that reviewer comments are in black and our response is in green. Per request, we have also enclosed a revised manuscript with all changes tracked.

Sincerely,

Krishna Pagilla

Review 1

Comments and Suggestions for Authors

In this manuscript, the authors attempted to investigate the biomass density on both cell level and floc level, and hence provide the overall guidance based for process effects and control. Firstly, the author was established biomass density separation method, and then applied qPCR to quantify four target functional groups in different density layers; combining with FISH, the morphology of flocs influences density distribution. Finally, used ordination analysis to study the influence of operational conditions in different WRRFs on AS solids density. Overall, this study fit the aim and scope of ijerph, and can lead to good audience in the field of wastewater engineering.

The specific comments were:

Flocculation is important for solid-liquid separation of activated sludge and should be mentioned in the introduction.

Reponses: Following this comment, we have extended the introduction to include information regarding flocs and flocculation (page 1, line 41-65 in the revised manuscript). New citations have been added (ref 6 - 10 in the revised manuscript). Besides, we have largely revised the introduction to make it more concise and cohesive. We thank the reviewer for this comment.

In section 2.3. What is the housekeeping gene for Q-PCR in this study? How to ensure the uniformity of Q-PCR results?

Response: There are two qPCR methods, namely absolute quantification and relative quantification. In this study, we performed absolute quantification. Unlike relative quantification that requires simultaneously measuring the target gene and a reference gene (usually a housekeeping gene), absolute quantification uses PCR amplicons cloned on plasmids as positive standards to create calibration curves. To avoid potential confusion, we have moved relevant information from the Supplementary Materials to the main text (page 4, line 410-415 in the revised manuscript). We also reorganized the Supplementary Materials, which now starts with PCR amplification, followed by PCR Product Cleanup and Plasmid Analysis, and then qPCR.

Line 196. “by what”?

Response: Following the reviewer’s suggestion, we have revised the sentence (page 8, line 207). We thank the reviewer for careful reading.

Line 196-199, 215-217, et al. Please move the unrelated part to the section discussion.

Response: Following the reviewer’s suggestion, we have moved discussion contents from the Results section to the Discussion section (see Page 8, Line 208-209 and Page 12, Line 307-309 in the revised manuscript). However, we would argue that DNA yield (line 197-199 in the original manuscript) is direct result rather than discussion. Therefore, we would like to keep those contents in the Results section.

Line 225-230. Please show specific data.

Response: The floc size data were calculated from microscopic observations of 20 flocs for each WRRF. Example microscopic images for each WRRF have been provided in the Supplementary Materials (Page 12, Line 456 in the revised manuscript). We have also removed the phrase “data not shown” from the main text and revised the sentences accordingly (page 9, line 232-235).

In figure 3. Why the scale bar is not uniform.

Response: We agree with the reviewer that 1 of the 16 CLSM images has a larger scale bar compared to the other images (i.e., larger magnification). However, this should not affect the conclusions as all the scale bars indicate the same size of 20 mm. Additionally, we have added more CLSM images to the Supplementary Materials (see Figure S3 and S4).

In figure 4. Figure 4 is incomplete.

Response: We thank the reviewer for careful reading. Figure 4 has been revised. In addition, we re-performed db-RDA using enrichment factors (fold change in relative abundance) rather than relative abundance itself as enrichment factors consider both the original relative abundance and the relative abundance after separation/enrichment. Both the new Figure 4 and corresponding main text (page 13, line 361–367) reflect this change.

The section conclusion needs to be further streamlined, only highlighting the main conclusion.

Reponse: This comment has been taken. We have largely revised the Conclusions section to make it more concise and conclusive (page 14, line 377-387).

Line 360-367, et al. Move the outlook of this research to the section discussion.

Reponse: We have moved some of the discussion from the Conclusions section to the Discussion section (page 6, line 360-366 in the revised manuscript) while removed the remaining discussion from the Conclusions section.

In figure S1. Species names need to be italicized. (done)

Reponse: This comment has been addressed (page 16, Figure S1.).

Reviewer 2 Report

The Authors have performed density-based separation of microbial groups in activated sludge (AS). They have determined among other things that the highest enrichment of microbial groups was observed in the sludge fraction denser than 1.036 g/cm3 and fraction lighter than 1.030 g/cm3, depending on the AS origin.

The study is interesting and adheres to a very important issue, i.e. biological wastewater treatment. Moreover, it fits to the scope of the journal has good level of novelty and English. I can recommend it after addressing several minor comments.

„Several studies have been done on how bacteria affect average solids density in AS,” – citations should be provided. Sometimes in the manuscript, I saw slight confusion in the units e.g. mL and cm3 are provided in the eq 1. The units should be unified within the manuscript. “5MGD, millions of gallons” – this could be recalculated to dm3 or L. L209: g/cm3 – superscript missing.

Author Response

Review 2

Comments and Suggestions for Authors

The Authors have performed density-based separation of microbial groups in activated sludge (AS). They have determined among other things that the highest enrichment of microbial groups was observed in the sludge fraction denser than 1.036 g/cm3 and fraction lighter than 1.030 g/cm3, depending on the AS origin.

The study is interesting and adheres to a very important issue, i.e. biological wastewater treatment. Moreover, it fits to the scope of the journal has good level of novelty and English. I can recommend it after addressing several minor comments.

Several studies have been done on how bacteria affect average solids density in AS,” – citations should be provided.

Reponse: According to reviewer’s suggestion, we have added 3 new citations (page 2, line 55; ref 5, 12, 15 in the revised manuscript).

Sometimes in the manuscript, I saw slight confusion in the units e.g. mL and cm3 are provided in the eq 1. The units should be unified within the manuscript. “5MGD, millions of gallons” – this could be recalculated to dm3 or L.

Reponse: According to reviewer’s suggestion, we have changed units from MGD to m3/day and from mL/L to cm3 to dm3 when needed (page 3, Table 1). Some units are kept as mg/L (e.g., for MLVSS, BOD, total N, total P, TSS) due to their conventional use.

L209: g/cm3 – superscript missing. 

Reponse: This comment has been taken (page 8, line 219).

Reviewer 3 Report

The paper is very interesting and it worth to be published.

Minor suggestions are following:

1- line 14. Define the acronym 

2- line 64. Please, quote what studies you refer to;

3- line 151. Review font size

4 Fig.1 (right side). Please, consider change colours, as they are too close in tone.

5- Maybe it’s worth to mention how much is expected for AS flocs’ structure modification, as result of sampling methods, and how this may interfere on measurements.

6- Line 338. Cap letter for db-RDA

Author Response

Review 3

Comments and Suggestions for Authors

The paper is very interesting and it worth to be published.

Minor suggestions are following:

1- line 14. Define the acronym

Reponse: This comment has been taken. All acronyms are defined in the revised Abstract (page 1, line 8-23).

2- line 64. Please, quote what studies you refer to;

Reponses: According to reviewer’s suggestion, we have extended the sentence and added the citations (page 2, line 66).

3- line 151. Review font size

Reponse: We have revised the font size.

4 Fig.1 (right side). Please, consider change colours, as they are too close in tone.

Reponse: We have used different colors in the new Figure 1. Thanks for this comment.

5- Maybe it’s worth to mention how much is expected for AS flocs’ structure modification, as result of sampling methods, and how this may interfere on measurements.

Reponse: Following this suggestion, we have discussed how sampling procedure might have affected AS floc structure, especially influence of EPS on floc stability and AS flocculation. Please see page 5, line 341- 349.

6- Line 338. Cap letter for db-RDA

Reponse: This comment has been addressed.

Round 2

Reviewer 1 Report

The author has made the necessary revisions and the manuscript can be published now.